

# Impacts of Nitrogen Addition on Nitrous Oxide Emission: Model-Data Comparison

Yujin Zhang[1], Minna Ma[2], Huajun Fang[3], Dahe Qin[1], Shulan Cheng[4], and Wenping Yuan[1*]

[1] State Key Laboratory of Cryospheric Sciences, Northwest Institute of Eco-Environment and Resources, Chinese Academy of Sciences, Lanzhou, Gansu 730000, China
[2] School of Atmospheric Sciences, Sun Yat-Sen University, Zhuhai, Guangdong, 519082, China
[3] Key Laboratory of Ecosystem Network Observation and Modeling, Institute of Geographical Sciences and Natural Resources Research, Chinese Academy of Sciences, Beijing, China
[4] College of Resources and Environment, University of Chinese Academy of Sciences, Beijing 100049, China

*Corresponding author:* Wenping Yuan (wyuan@lzb.ac.cn)

**Abstract.**

The contributions of long-lived nitrous oxide ($N_2O$) to the global climate and environment have received increasing attention. Especially, atmospheric nitrogen (N) deposition has substantially increased in recent decades due to extensive use of fossil fuels in industry, which strongly stimulates the $N_2O$ emissions of the terrestrial ecosystem. Several models have been developed to simulate $N_2O$ emission, but there are still large differences in their $N_2O$ emission simulations and responses to atmospheric deposition over global or regional scales. Using observations from N addition experiments in a subtropical forest, this study compared six widely-used $N_2O$ models (i.e. DayCENT, DLEM, DNDC, DyN, NOE, and NGAS) to investigate their performances for reproducing $N_2O$ emission, and especially the impacts of two types of N additions (i.e. ammonium and nitrate: $NH_4^+$ and $NO_3^-$, respectively) and two levels (low and high) on $N_2O$ emission. In general, the six models reproduced the seasonal variations of $N_2O$ emission, but failed to reproduce relatively larger $N_2O$ emissions due to $NH_4^+$ compared to $NO_3^-$ additions. Few models indicated larger $N_2O$ emission under high N addition levels for both $NH_4^+$ and $NO_3^-$. Moreover, there were substantial model differences for simulating the ratios of $N_2O$ emission from nitrification and denitrification processes due to disagreements in model structures and algorithms. This analysis highlights the need to improve representation of $N_2O$ production and diffusion, and the control of soil water-filled pore space on these processes in order to simulate the impacts of N deposition on $N_2O$ emission.

**Keywords:** Nitrous oxide; Model; Nitrogen deposition; Forest; Model-data comparison






## 1 Introduction

Nitrous oxide (N$_2$O) is one of the most important greenhouse gases, and contributes 6.24 % to overall global radiative forcing as the third contributor after carbon dioxide and methane (Forster et al., 2007; WMO, 2011). N$_2$O plays an important role in depleting stratospheric ozone, which decreases harmful ultra-violet radiation reaching the earth. A doubling of the 35 atmospheric N$_2$O concentration could decrease the ozone layer by 10 % (Crutzen and Ehhalt, 1977; Ravishankara et al., 2009). Since the industrial revolution, the atmospheric N$_2$O concentration has increased nearly 21 % from about 270 ppbv during the pre-industrial era to 325.9 ppbv in 2013, with an average increase rate of about 0.82 ppbv yr$^{-1}$ during the last decade (WMO, 2014). Terrestrial ecosystems can act as either sources or sinks for atmospheric N$_2$O, depending on time and location (Potter et al., 1997; Ridgwell et al., 1999; Chapuis-Lardy et al., 2007; Xu et al., 2008). Globally, natural sources 40 from terrestrial ecosystems contribute more than 50 % to the N$_2$O releases to the atmosphere (Denman et al., 2007). Quantifying N$_2$O fluxes in global terrestrial ecosystems, therefore, is an urgent task for predicting future climate change (Sheldon and Barnhart, 2009).

Several process–based N cycle models have been developed and widely used for quantifying the spatial–temporal variations in N$_2$O flux (Li et al., 1992; Engel and Prentice, 1993; Grant et al., 1993; Potter et al., 1996; Xu and Prentice, 45 2008; Zhuang et al., 2012). In general, these models usually integrate key biogeochemical processes, including nutrient mineralization, immobilization, nitrification, and denitrification. However, there exist substantial model disagreements in the estimated magnitude and spatial distribution of N$_2$O at regional and global scales (Figure 1). For example, Xu and Prentice (2008) used the DyN model to estimate global terrestrial ecosystem N$_2$O emission at 13.31 Tg N yr$^{-1}$, which is 3.94 times the estimate of 3.37 Tg N yr$^{-1}$ arrived at by Zhuang et al. (2012) (Figure 1).

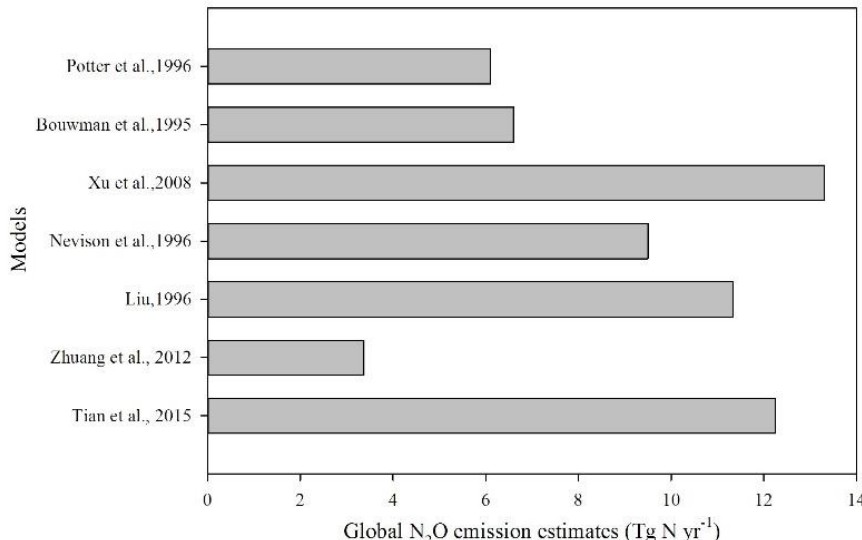

**Figure 1.** Comparison of global estimates of N$_2$O emission from the terrestrial ecosystem.




Each model is a combination of equations describing environmental regulations of $N_2O$ emission. Individual model validations, however, are not sufficient to identify the sources of the wide range of model differences. A rigorous comparison must be conducted in a standardized framework with consistent validation datasets and driving variables. To

generate more robust estimates of $N_2O$ flux dynamics, it is necessary to compare estimates from a variety of $N_2O$ models and compare them against consistent and extensive measurements that are available.

Atmospheric nitrogen (N) deposition, which is closely related to $N_2O$ emission, has shown a strong increasing trend in recent decades due to the extensive use of fossil fuels in industry and transportation and the heavy application of fertilizers in agriculture (Galloway et al., 2004). It is estimated that global atmospheric N deposition has increased from 1 Tg N in the

1860s to 25–40 Tg N in the 2000s, and is projected to continuously increase to 210 Tg N by the year 2050 (Neff et al., 2002; Lamarque et al., 2005; Galloway et al., 2008; Lu et al., 2016). The forest ecosystem in eastern China was recognized as the region receiving the highest atmospheric N deposition in southeast China (Liu et al., 2013). The N deposition input into terrestrial ecosystems alters plant physiology and the soil microbial community (Litten et al., 2007; Treseder, 2008), thereby changing the soil biogenic $N_2O$ flux (Butterbach–Bahl, 1997; Allen et al., 2004; Bange, 2006; Chen et al., 2015). Based on a

meta–analysis of N addition experimental data worldwide, Liu and Greaver (2009) concluded that N addition could increase $N_2O$ emission by up to 216 %. In general, chronic N deposition will increase ammonium ($NH_4^+$) and nitrate ($NO_3^-$) availability in terrestrial ecosystems, thereby affecting $N_2O$ flux through changing the activity and composition of the microbial community (Smith et al., 2003; Bowden et al., 2004; Monteny et al., 2006). However, to our knowledge, few studies have evaluated model performance in simulating the impacts of N deposition on $N_2O$ emission.

In this study, six widely–used $N_2O$ models, i.e. DayCENT (the daily version of the CENTURY ecosystem model; Parton et al., 1996, 2001; Del Grosso et al., 2001), DNDC (the Denitrification–Decomposition model; Li et al., 2000), DLEM (Dynamic Land Ecosystem Model; Tian et al., 2010), DyN (the global Dynamic Nitrogen model; Xu and Prentice, 2008), NOE (the algorithm of Nitrous Oxide Emission; Henault et al., 2005), and NGAS (a hybrid of a process–oriented model and a nutrient cycling model; Parton et al., 1996), were chosen for examination of their performance under different

levels of N deposition in a subtropical forest in southeast China. The study aims to (i) examine performance of the models in a forest ecosystem, particularly for seasonal variations of $N_2O$ emission, (ii) investigate the ability of these models under different levels of N deposition as well as two N types, and (iii) identify the key issues in the application of these models and future model development so as to improve their simulation of $N_2O$ emissions.

## 2 Materials and Methods

### 2.1 Study Site

This model–data comparison is based on field experiments conducted at a subtropical evergreen forest station, the Qianyanzhou Ecological Station (26°44′39″ N, 115°03′33″ E). The station is in Jiangxi Province of southern China, which is one of the important regions subject to atmospheric N deposition. The study plots were located in the slash pine plantation



established in 1958. Average tree height was about 15 m, with diameter at breast height of 16.1 cm, stand basal area of 35 m$^2$ ha$^{-1}$, and leaf area index of 4.5. Dominant understory and midstory species are *Woodwardia japonica* (L.f.) Sm., *Dicranopteris dichotoma* (Thunb.) Bernh., *Loropetalum chinense* (R.Br.) Olv., and *Quercus fabrei* Hance. The typical soil is weathered from red sandstone and mud stone. Soil texture is divided into 2.0–0.05 mm (17 %), 0.05–0.002 mm (68 %), and < 0.002 mm (15 %). Soil bulk density, organic carbon, total N content, and pH of the surface part (0–40 cm) were 1.57 g cm$^{-3}$, 7.2 g kg$^{-1}$, 0.55 g kg$^{-1}$, and 4.6, respectively. The study site has a humid monsoon climate with a mean air temperature of 17.9 ℃ and precipitation of 1469 mm per year. A large portion of the precipitation occurs in spring and early summer, but it is relatively dry in late summer and autumn with high air temperatures and low precipitation.

## 2.2. Field Experiments

The field experiments were conducted during April–December 2012. According to previously reported levels of atmospheric N deposition at the study area (Wang et al., 2011), two levels (low and high N of 0 and 120 kg N ha$^{-1}$ yr$^{-1}$, respectively) of two different N fertilizers (NH$_4$Cl and NaNO$_3$) were applied to mimic two future scenarios of N deposition. At the same time, a control experiment was carried out for comparison. Each level of N treatment was conducted in a plot of 20 m × 20 m with a space of 10 m between any two plots. The N fertilizer solutions were sprayed on the plots once a month in 12 equal applications, and the control plots received only equivalent deionized water.

Flux data of N$_2$O were determined using a static opaque chamber and gas chromatography method (Fang et al., 2014), which were installed near an eddy covariance tower in the ecological station. Daily fluxes were collected from the measurements approximately every two weeks. The soil fluxes were calculated based on the rate of changes in their concentration within the chamber, estimated as the slope of the linear regression between concentration and time (Wang et al., 2011). Soil temperature at 5 and 10 cm depths were monitored at each chamber site, using portable temperature probes (JM624 digital thermometer, Living–Jinming Ltd., Tianjin, China). At the same time, soil samples were collected nearby the static chambers from a depth of 0–20 cm using an auger (2.5 cm in diameter). Volumetric soil moisture (m$^3$ m$^{-3}$) was measured using a moisture probe meter (TDR100, Spectrum Technologies Inc., PlainField, IL, USA). Soil pH was also measured using the potentiometry method. Soil water–filled pore space (WFPS) was calculated using the methods reported by Fang et al. (2014).

## 2.3. N$_2$O Models

Six N$_2$O models were selected in this model-data comparison: DayCENT (Parton et al., 1996, 2001; Del Grosso et al., 2001), DNDC (Li et al., 2000), DLEM (Tian et al., 2010), DyN (Xu and Prentice, 2008), NOE (Henault et al., 2005), and NGAS (Parton et al., 1996). All six investigated N$_2$O models are based on two major microbial processes: nitrification and denitrification, which are separately simulated from these two processes using the following equation:

$$F_{N_2O} = F_{nt} + F_{dn} \tag{1}$$



where $F_{N_2O}$ is the N$_2$O emission from soil to air (g N m$^{-2}$ day$^{-1}$), and $F_{nt}$ and $F_{dn}$ are N$_2$O emissions from nitrification and denitrification processes, respectively. Detailed model algorithms can be found from the Supplemental Online Materials.

### 2.4. Simulation Protocol and Parameter Inversion

The field observations of soil temperature, soil moisture, pH, soil respiration, dissolved organic carbon, soil NH$_4^+$ content, and soil NO$_3^-$ content were used to drive the six models. As one of the key drivers, WFPS was derived using the

following equation (Fang et al., 2014):

$$WFPS = VWC/(1 - BD/2.65) \qquad (2)$$

where $VWC$ is soil volumetric moisture content (%), $BD$ is soil bulk density (g cm$^{-3}$), and 2.65 is soil particle density (g cm$^{-3}$).

The nonlinear regression procedure (Proc NLIN) in the Statistical Analysis System (SAS, SAS Institute Inc., Cary,

NC, USA) was applied to optimize the model parameters using observed N$_2$O emission for all five experiments. The calibrated parameter values were used to simulate N$_2$O emissions (Table S1).

Three metrics were used to evaluate the performance of these models:

(i) The coefficient of determination between observation and simulation (R$^2$).

(ii) Absolute predictive error (PE), quantifying the difference between simulated and observed values.

(iii) Relative predictive error (RPE), computed as:

$$RPE = (\bar{S} - \bar{O})/\bar{O} \times 100 \qquad (3)$$

where $\bar{S}$ and $\bar{O}$ are mean simulated and mean observed values, respectively.

### 3 Results

All six models generally reproduced the seasonal variations of measured N$_2$O fluxes for the control and four N

addition experiments. The measurements showed the largest N$_2$O emissions during April–July, and the lowest in winter (Figure 2). The simulated emissions showed some differences in estimates for various models. Although the simulated N$_2$O emissions from different models decreased from spring and summer to autumn and winter, indicating the seasonal pattern of emissions (Figure 1), there were some abrupt changes in model estimates. Most models captured the peak and trough of N$_2$O emission. Collectively, the six models explained 1 %−16 % of the variations in N$_2$O fluxes across all experiment plots (Table

140 1).





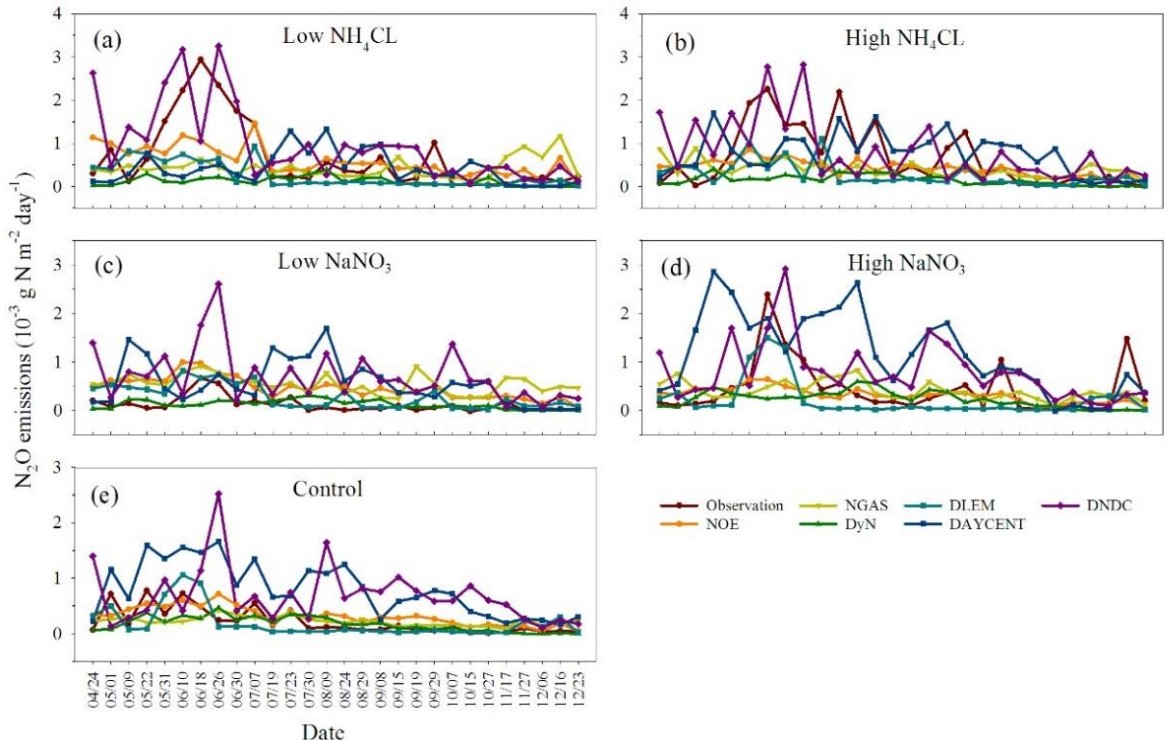

**Figure 2.** Comparisons of N₂O emission simulations and observations for five experiment treatments.

Most models did not fully indicate the stimulations of N additions to N₂O emission that were observed in field experiments. According to the observed N₂O fluxes, $NO_3^-$ and $NH_4^+$ additions increased N₂O emission for four addition experiments, and high $NO_3^-$ and $NH_4^+$ additions led to higher N₂O emission compared to low additions (Figure 3). Furthermore, larger increases of N₂O emission occurred for $NH_4^+$– compared to $NO_3^-$–addition experiments (Figure 3). However, NGAS, DyN, DayCENT, and DNDC models simulated larger $NO_2$ fluxes for low compared to high $NH_4^+$– addition treatments (Figure 3). NOE and NGAS did not correctly indicate the differences of N₂O fluxes between high and low $NO_3^-$ treatments. In addition, the experiments also indicated higher simulations of N₂O emission for $NH_4^+$ compared with $NO_3^-$ additions. However, only NOE and DLEM models reproduced larger impacts of $NH_4^+$ on N₂O emissions compared with low $NH_4^+$ level.

**Table 1**. Predictions of the six N₂O models for four N addition treatments

|  | Model/N level | Low NH₄Cl | High NH₄Cl | Low NaNO₃ | High NaNO₃ | Control |
|---|---|---|---|---|---|---|
| Obs | Mean# | 0.69[b] | 1.14[a] | 0.29[c] | 0.78[ab] | 0.18[c] |
| NOE | Mean | 0.62[ab] | 0.62[b] | 0.53[b] | 0.48[b] | 0.47[a] |
|  | R² | 0.55* | 0.35* | 0.61* | 0.40* | 0.33* |
|  | PE | −0.07 | −0.51 | 0.24 | −0.30 | 0.29 |
|  | RPE (%) | −11.16 | −44.92 | 84.45 | −38.43 | 163.51 |




| | | | | | | |
|---|---|---|---|---|---|---|
| NGAS | Mean | 0.55$^b$ | 0.29$^d$ | 0.49$^b$ | 0.35$^d$ | 0.22$^{bc}$ |
| | $R^2$ | 0.11 | 0.27$^*$ | 0.50$^*$ | 0.01 | 0.14$^*$ |
| | PE | −0.16 | −0.85 | 0.19 | −0.42 | 0.042 |
| | RPE (%) | −22.33 | −74.52 | 69.47 | −54.73 | 24.33 |
| DyN | Mean | 0.49$^b$ | 0.41$^c$ | 0.48$^{bc}$ | 0.58$^b$ | 0.43$^{ab}$ |
| | $R^2$ | 0.28$^*$ | 0.23$^*$ | 0.14$^*$ | 0.15$^*$ | 0.21$^*$ |
| | PE | −0.21 | −0.73 | 0.19 | −0.20 | 0.28 |
| | RPE (%) | −29.65 | −64.92 | 67.11 | −26.02 | 139.70 |
| DLEM | Mean | 0.51$^b$ | 0.79$^{ab}$ | 0.42$^{bc}$ | 0.43$^b$ | 0.29$^b$ |
| | $R^2$ | 0.38$^*$ | 0.24$^*$ | 0.58$^*$ | 0.79$^*$ | 0.57$^*$ |
| | PE | −0.19 | −0.35 | 0.14 | −0.39 | 0.11 |
| | RPE (%) | −27.46 | −30.87 | 47.86 | −44.51 | 64.15 |
| DayCENT | Mean | 0.61$^{ab}$ | 0.52$^a$ | 0.62$^{ab}$ | 0.98$^a$ | 0.51$^a$ |
| | $R^2$ | 0.13$^*$ | 0.14$^*$ | 0.01 | 0.23$^*$ | 0.28$^*$ |
| | PE | −0.09 | −0.62 | 0.33 | 0.20 | 0.34 |
| | RPE (%) | −12.16 | −54.51 | 115.36 | 24.92 | 188.84 |
| DNDC | Mean | 0.92$^a$ | 0.78$^{ab}$ | 0.78$^a$ | 0.98$^a$ | 0.59$^a$ |
| | $R^2$ | 0.35$^*$ | 0.13$^*$ | 0.25$^*$ | 0.31$^*$ | 0.01 |
| | PE | 0.22 | −0.36 | 0.49 | 0.19 | 0.41 |
| | RPE (%) | 30.85 | −31.94 | 172.51 | 24.80 | 234.71 |

*Note.* Letters indicate significant differences among N$_2$O values for the same levels of N addition from different model simulations or observation. $R^2$ is the coefficient of determination between observation and simulation. PE is absolute predictive error. RPE is relative predictive error. # indicates the mean value of observed or simulated N$_2$O emissions ($10^{-3}$ g N m$^{-2}$ day$^{-1}$). * indicates the significance of $p < 0.05$.





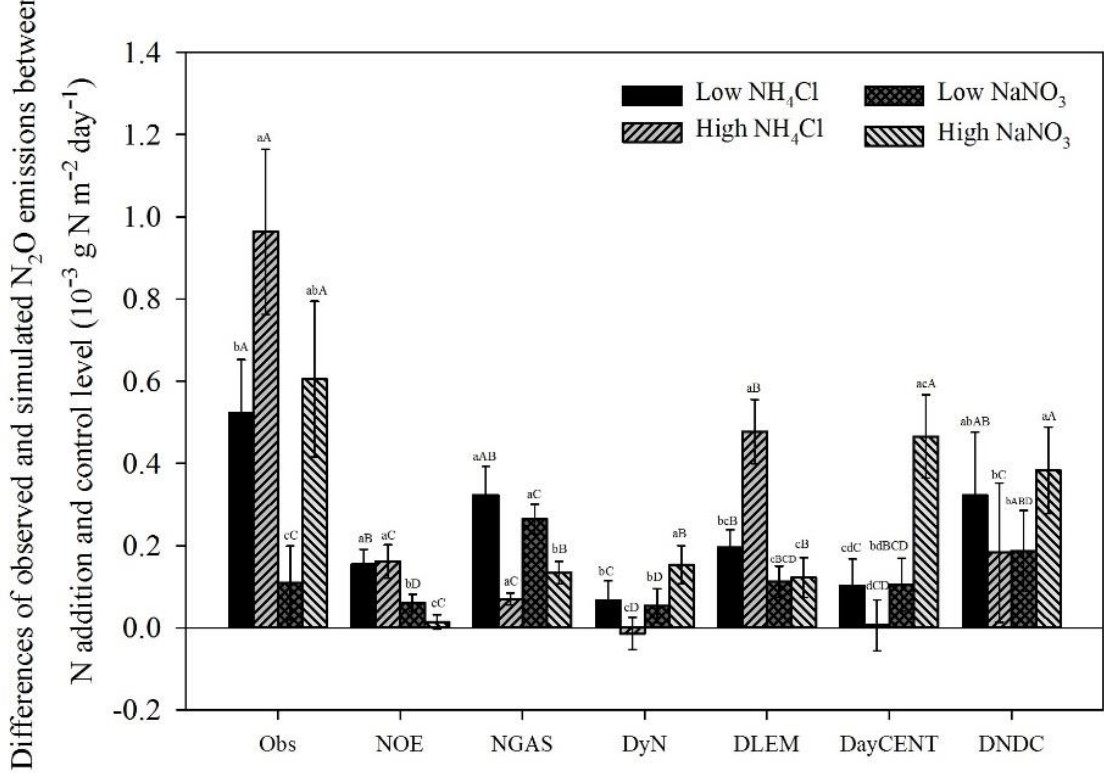

**Figure 3.** Comparisons of N₂O emission differences between N addition and control treatments from observation and model simulations. Lower–case letters indicate significant differences among the values for different N addition level for an individual model or observation. Capital letters indicate the significant difference among the values for the same levels of N addition from different model simulations or observation.

Because N₂O emissions are generally from two different microbial processes, i.e. nitrification and denitrification, the proportions of N₂O emissions due to both processes were calculated to quantify their contributions to total emissions. All six models showed consistently negative correlations between the ratios of N₂O emission from nitrification and WFPS (Figure 4a). The six models showed that nitrification contributed more than half of N₂O emissions; however, there were large differences in the ratios of N₂O fluxes generated by nitrification and denitrification among the models (Figure 4b). On average, the DayCENT model simulated the lowest ratio (about 55.4 %) of N₂O emissions generated by nitrification, and the largest ratio (about 89.5 %) was for the DyN model (Figure 4b).



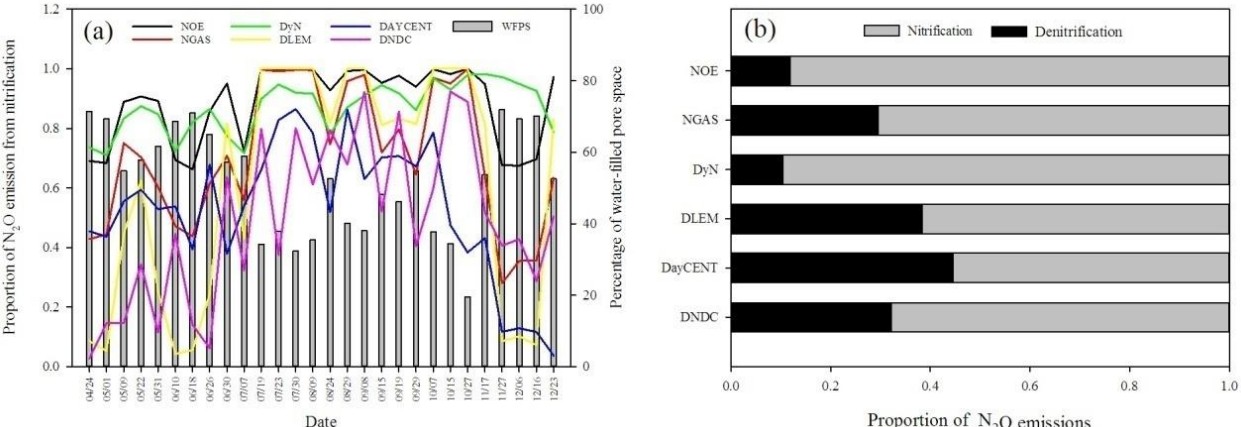

**Figure 4.** Proportions of $N_2O$ emissions for different models. (a) Seasonal proportion of $N_2O$ emission from nitrification and seasonal water–filled pore space (WFPS). (B) Averaged proportion of emissions from nitrification and denitrification.

## 4 Discussion

### 4.1. Model Performance

Compared with ecosystem carbon dioxide emissions, few studies have evaluated model performance for simulating $N_2O$ emissions due to the relative scarcity of N cycle measurements (Henault et al., 2012). Notably, N depositions from the atmosphere have been documented to increase with industry processes (Galloway et al., 2004), which are believed to have significant impacts on soil $N_2O$ emissions due to their impacts on microbial processes. Therefore, the sixth IPCC report, which will be conducted in the next five years, requires Earth System Models integrate N cycle (IPCC, 2017). Therefore, process–based $N_2O$ models have been widely developed and applied in recent years (Li et al., 1992; Engel and Priesack, 1993; Parton et al., 1996; Potter et al., 1997; Del Grosso et al., 2001; Henault et al., 2005; Xu and Prentice, 2008; Tian et al., 2010). These models are now being used not only for the prediction of $N_2O$ emissions from different ecosystems, but estimation of $N_2O$ inventories on national, regional, and global scales, and for assessing climate change impacts and mitigation strategies (Del Grosso et al., 2006, 2009; EPA, 2006). However, it should be noted that these model predictions may not be reliable when applied to a new environment, and their performance should be first tested with different data streams from real world experiments.

Our comparison showed the general performance of six investigated models in reproducing seasonal variations and magnitudes of $N_2O$ emissions (Figure 2). This conclusion was supported by several recent model evaluations, which revealed unstable performance of $N_2O$ models (Senapati et al., 2016). For example, different studies with the DayCent model have found a range of correlations from weak to strong across different agroecosystems (Henault et al., 2012). Parton et al. (2001) found correlations between daily measured vs. simulated $N_2O$ emissions, with range 0–0.44, from a variety of five



different grassland sites in the USA. Other ecosystem models also face similar difficulties in simulation of daily $N_2O$ emissions, for example DNDC (Yeluripati et al., 2015) and CoupModel (He et al., 2016).

The comparison revealed the complexity in modeling the impacts of N addition on $N_2O$ emission. The field observations in the current study indicated larger $N_2O$ emissions for $NH_4^+$ compared with $NO_3^-$ additions at two addition

levels (Figure 3). However, these impacts were not reproduced by all of the six models except for the DELM and NOE models. Previous study showed that the impacts of $NH_4^+$ addition on $N_2O$ emissions are, to some extent, larger compared with $NO_3^-$ addition (Wang et al., 2016). This is probably due to two primary reasons. One is that under favorable temperature and moisture, nitrification dominates $N_2O$ emission compared with denitrification if soil is acidic and rich in $NH_4^+$. The addition of $NH_4^+$ can significantly increase the substrates for ammonia–oxidizers and the abundance of ammonia–oxidizing

archaea, which give rise to increases in soil autotrophic nitrification rate (Gao et al., 2016a; 2016b). The other reason is that additions of $NH_4^+$ fertilizers can have larger impacts on the acidification of soil compared with the additions of $NO_3^-$, which is closely related to the accumulation of $H^+$ in soil solution and the leaching of $NO_3^-$ from soil (Tian and Niu, 2015). Soil acidification decreases availability of $NH_4^+$, which is favorable to the growth of soil nitrifiers, i.e. ammonia–oxidizing archaea, but unfavorable to soil denitrifiers (Isobe et al., 2012).

**4.2. Structure Differences among $N_2O$ Models**

The performance of these $N_2O$ models strongly depends on model algorithms, and also on the major pathways of $N_2O$ emissions and their responses to environmental conditions. This is because the processes of $N_2O$ emissions are extremely competitive and are controlled by many drivers, e.g. soil temperature, moisture, soil redox potential, and the availability of substrates for microbes (Schmidt et al., 2000). In the present study, the models did not adequately capture the environmental

regulation of $N_2O$ emission. Nitrification and denitrification are two major processes of $N_2O$ production. Numerous experiments have shown that nitrification and denitrification can occur simultaneously because of the coexistence of aerobic and anaerobic zones in soils (Henault et al., 2012; Hu et al., 2015); however, the availability of soil oxygen–determined by soil water content and other soil properties – strongly regulates the proportion of nitrification and denitrification (Li et al., 1992). Numerous studies have investigated the relationship between soil moisture and the contributions of nitrification and

denitrification processes. In N fertilizer–amended soil, $N_2O$ emission has been found to be highly correlated with WFPS, with the highest emission at around 70 % WFPS, which was attributed to a combination of nitrification (35 %–53 %) and denitrification (only 2 %–9 %) (Huang et al., 2014). In sandy loam soils, when moisture status was sub–optimal for denitrification (50 % and 70 % WFPS), nitrification was the significant contributor (around 29 %) to $N_2O$ emissions (Kool et al., 2011); however, in wetter soils (–0.1 kPa) nitrification contributed less than 3 % (Webster and Hopkins, 1996). Well et al.

(2008) attributed 88 % of total $N_2O$ emission to nitrification at 45 % WFPS. This suggests that favorable conditions for $N_2O$ production from nitrification occur within the range of 30 %–70 %, whereas denitrification dominates $N_2O$ production in wet soils with WFPS>80 % (Braker and Conrad, 2011; Huang et al., 2014). The values of WFPS in the current study were within the range of 30 %–70 %, which was favorable to the occurrence of nitrification in all of these models.

In general, all six N$_2$O models use soil water content to control the balance of two processes. NOE uses a simplified
25 scheme to separate the nitrification and denitrification processes. Nitrification only occurs if WFPS < 80 %, whereas
denitrification only occurs if WFPS > 62 %; within the range of 62 %−80 %, the two processes may occur simultaneously
(Henault et al., 2005). For DLEM, denitrification and nitrification are simulated as a one−step process. Due to the effect of
soil moisture, denitrification only occurs when soil moisture exceeds field capacity (Tian et al., 2010). For DyN and DNDC,
aerobic and anaerobic microsites are assumed to simultaneously exist in most soils. Nitrification occurs in aerobic microsites,
but denitrification is mainly in anaerobic microsites. The key factor affecting the ratio between aerobic and anaerobic
microsites is soil redox potential, which controls the ratio between nitrification and denitrification (Li et al., 1992; Xu and
Prentice, 2008). For NGAS and DayCENT, no specific threshold is applied for the occurrences of the two processes and they
are assumed to occur simultaneously (Parton et al., 1996, 2001; Del Grosso et al., 2001). Thus, the differences in the
algorithms of the six models are believed to be the key reasons for the differences in the model estimates of N$_2$O emission.

**5 Conclusions**

We examined the performance of six N$_2$O models for indicating the impacts of different levels of N addition on N$_2$O
emission. Results indicated that the investigated models can represent the general seasonal variations of N$_2$O emissions
under both N addition and non−N addition levels. However, additions of NH$_4^+$ rather than NO$_3^-$ could have more significant
effects on N$_2$O emissions from soils, which were not represented by most of the models. In addition, most of the models
failed to reproduce larger N$_2$O emissions at high level of nitrate additions compared with ammonia additions. Moreover, the
analysis suggested that the disagreements in model structure and algorithms resulted in substantial differences in N$_2$O
emission and mediating processes (i.e. nitrification and denitrification).

*Competing interests*. The authors declare that they have no conflict of interest.


*Data availability*. The data can be obtained upon request to the authors.

**Acknowledgments**

This study was supported by the National Key Basic Research Program of China (2016YFA0602701), the Major
Programs of High−Resolution Earth Observation System (Grant No. 32−Y2−0A17−9001−15/17), the One Hundred Person
Project of CAS, the National Youth Top−Notch Talent Support Program (2015−48), and the Youth Changjiang Scholars
Programme of China (Q2016161). We thank many members of the filed crew who collected the field data for the study. We
also thank the anonymous reviewers for their comments.





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
