# Peer review of "Impacts of Nitrogen Addition on Nitrous Oxide Emission: Model-Data Comparison"

_Biogeosciences, 2018_

## Referee Comment (RC1) · Anonymous Referee #1 · 26 Apr 2018

General consideration: This manuscript presents a model intercomparison for assessing models response to N2O emissions using different level of N deposition in a subtropical forest in China. Despite the huge potentiality of the work, mainly thanks to the availability of instruments which allows to retrieve information on several control factors of process-based models, the global work is not well structured. The paper does not fluent and the explanations of all the modelling aspects (i.e. initialization, sensitivity, calibration and validation) are missing. Also, English is poor. I recommend rejection for this paper.

Critical points

There are many critical points which were here briefly reported. In general, the paper lack of all basic aspects which should be considered in a publication focused on

models.

1. The paper does not provide any formation about how models have been initialized: What about model spin-up (pools' equilibrium)? Which are the main input parameters (climate, soil, vegetation and management?)? What about climate (how the ini file is structured? what does it requires?)? From where these climate info were retrieved? ? What about soil? From where these soil info were retrieved? Do exists a meteorological station? Far or close to the experimental area?.

2. The paper does not provide any formation about how the model has been calibrated. Authors cannot only apply models. This option could be possible only if they are able to provide a strong background for each model. This can be done, however, only for the most commonly applied tools (i.e. DNDC and DayCent, but very difficult for the remaining). Authors have to be able to provide proofs that models are able to reproduce a specific ecosystem. To do that, they should calibrate the model, reporting results related to fluxes but also to biomass or other parameters (i.e. SOC dynamics, fruit pools, etc). Doing so, authors would proof that their models are able to reproduce all these parameters. Otherwise the different effect that a specific vegetation type (as example oaks or pine for forest, maize or rice for crop, warm or cold grass etc.) may play on N emissions is ignored. If this effect is ignored, this is means that models are not reproducing what does really happening in the field.

3. I don't think all these models are able to reproduce forestry systems. For instance, authors talk about DNDC. But does exist a specific version for simulating forest dynamics (namely forest DNDC). The common version (the latest one is DNDC95) is not appropriate for simulating forest systems. If authors have used this latter, they should explain much in detail what they do for simulating a forest system (i.e. how has been calculated biomass partitioning? From where they retrieved biomass partitioning info?).

4. The paper does not provides any formation about ecophysiological parameters used for reproducing a specific type of tree. Trees have different response to climate (i.e.

xenophile species, different RUE and WUE, different root depth and length). All these characteristics affect N amount and its permanence in the soil. In this paper is not possible to understand if these info were used (how they were partitioned within the model?) and how were retrieved (i.e. literature? Experiments?).

5. There are no information related to climate scenarios. Authors write about about future scenarios of N deposition but they do not refer to any climate scenarios (SRES? RCP? Which scenarios? Which time slice? References?).

6. Only one year of data is not enough for representing fluxes dynamics, especially considering that fluxes variability is closely related to climate-soil interaction. More than one year of data is needed for proving that models well work. This time should be used for calibrating the different models.

7. Figures are not clear and discussion is very poor.

---

## Referee Comment (RC2) · Anonymous Referee #2 · 14 May 2018

While the topic is relevant and within scope of the journal, the work itself is a combination of either sketchily presented or just plain poorly conducted and I recommend rejection. Some (but by no means all points) include: - The soil description is in no way sufficient for the work the be replicated (as is the scientific standard) - 120 kg N /ha /yr of atmospheric deposition? And if 0 kg N /ha /yr was a treatment level of N input then what was the control? - The presentation of the data handling was vastly inadequate such that a proper review was not possible - No information on model set up, how differences in basic assumptions about soils, input parameters etc. between models were handled - How were the forest systems set up in each model? How were they initialised and/or spin up? - Figures presented such that there were no uncertainties on the data given and it is not practically possible to differentiate the measurements from

the modelled outputs in the figures I see little prospect of this work being improved to the point that it would be publishable and recommend rejection from this journal.

---

## Referee Comment (RC3) · Anonymous Referee #3 · 16 May 2018

The aim of this paper is to investigate the performance of biogeochemical models (of different complexity) on nitrate and ammonium additions in terms of N2O emissions, mimicking the seasonal N depositions over a forest. I found this paper poor and lacking on multiple aspects, from the method to the discussion of the results.

Major points:

1. The calibration procedure of these models is not well reported in the text whilst it represent a pillar in this research field (section 2.4). I assume that no one of these models was previously calibrated, since most of them have not the possibility to simulate very complex systems as forest, neither plants. If this procedure has been performed, I would suggest the authors to detail it for another submission.

[Figure]

2. The models selected by the Authors present large differences in complexity and are constitutionally different. This could produce difficulties in using and comparing these models since some of them are just a module and are not able to consider important processes as the plant uptake, water dynamics in the soil, interaction with the biogeochemical cycle of C, or other losses of N than N2O as ammonia, as reported in the conclusions, but never discussed in the paper.

3. Some of the selected models is already set to add automatically the atmospheric deposition as a source of N to the system (wet and dry). A specific treatment by the Authors regarding the parameterization of these models to reduce this default addition were not addressed or discussed.

4. The title is not appropriate, since it does not circumscribe the domain of the investigation and does not uses specific words.

5. The abstract is very confusing and not well written in English, seems very different from other part of the text that appear much better written.

6. Introduction is lacking and not well shaped, related to global estimations, when the study in on another scale. Figure 1 that report only some example and is not exhaustive.

7. Materials and method par has to be improved, since there is not a clear explanation of the experiment, e.g. the role of the control plot, the repetitions (number and disposition), the measurement performed by the flux tower and the characteristics of the chambers, or soil depth of the measurements. In the simulation protocol there is no a detailed procedure on how the statistical procedure was applied.

8. Discussion section should be focused with the interpretation of the data, objectively and inter-subjectively, in light of the evidences brought by other scientists or backgrounds. In this section, there are part that are more suitable for the introduction section (i.e. L174-185). Some discussion is not directly in support of the obtained results.

Furthermore, the discussions are poor and are not addressing the aims of the paper.

9. Supplementary results. I appreciated the efforts of the Authors to collect the equations behind the N2O emissions in the explored models, but in some of them there are relations that have nothing to do with it e.g., Rg and Rd in DNDC. These equations and the parameters or variables used by each model could be the base to discuss the performances of each model.

10. Furthermore, I really suggest rephrasing everything avoiding plagiarism (that is at 25%).